



# C-GLORSv5: an improved multi-purpose global ocean eddy-permitting physical reanalysis

Andrea Storto[1], Simona Masina[1,2]

[1]Centro Euro-Mediterraneo sui Cambiamenti Climatici, Bologna, I-40128, Italy
[2]Istituto Nazionale Geofisica e Vulcanologia, Bologna, I-40128, Italy

*Correspondence to*: A. Storto (andrea.storto@cmcc.it)

**Abstract.** Global ocean reanalyses combine in-situ and satellite ocean observations with a general circulation ocean model to estimate the time-evolving state of the ocean, and they represent a valuable tool for a variety of applications, ranging from climate monitoring and process studies to downstream applications, initialization of long-range forecasts and regional

studies. The purpose of this paper is to document the recent upgrade of C-GLORS (version 5), a state-of-the-art ocean reanalysis produced at the Centro Euro-Mediterraneo per i Cambiamenti Climatici that covers the meteorological satellite era (1980-present) and it is being updated in delayed time mode. The reanalysis is run at eddy-permitting resolution (1/4 degree horizontal resolution and 50 vertical levels) and consists of a three-dimensional variational data assimilation system, a surface nudging and a bias correction scheme. With respect to the previous version (v4), C-GLORSv5 contains a number of

improvements. In particular, background- and observation- error covariances have been retuned, allowing a flow-dependent inflation in the globally averaged background-error variance. An additional constraint on the sea-ice thickness was introduced, leading to a realistic ice volume evolution. Finally, the bias correction scheme and the initialization strategy were retuned. Results document that the new reanalysis outperforms the previous version, especially in representing the variability of global heat content and associated steric sea level, the upper ocean temperature and the thermohaline circulation. The

dataset is available in NetCDF format at doi:**10.1594/PANGAEA.857995.**

## 1 Introduction

Ocean retrospective analyses (or simply reanalyses, or ocean syntheses) combine available in-situ and satellite observations with a ocean general circulation model (OGCM) forced by atmospheric reanalyses by means of advanced data assimilation techniques, with the aim of estimating the state of the ocean in the past decades (Balmaseda et al., 2015). Such a method

implicitly ingests not only the information coming from the global ocean observing system, but also the global meteorological observing system through the atmospheric forcing, and our knowledge of the ocean physics and dynamics through the OGCM.

Production of ocean reanalyses has been originally fostered by the need of monitoring the climate change signature on the ocean, for instance in terms of heat content evolution (Carton and Santorelli, 2008). Additionally, long-term (e.g. seasonal,



decadal) prediction systems require proper initial conditions for the ocean, because of the dominant effect of the ocean initial conditions on the predictability over long time scales (e.g. Balmaseda and Anderson, 2009). For this reason, ocean reanalyses are a crucial requirement for climate predictions, which further encourages their improved use of observations along time, and they can be produced either in delayed or near real-time mode. The former mode refers to the production and

update process that is generally performed occasionally when all reprocessed input observation datasets are available, while the latter refers to a continuous update process, possibly using real-time observations, generally adopted when the reanalysis serves the purpose of initializing an operational long-range prediction system (e.g. Zuo et al., 2015).

During the last decade, the maturity of ocean reanalyses has been shown through many validation studies (Stammer et al., 2010; Xue et al., 2012; Storto et al., 2015), which supported the idea that reanalyses provide useful information for the upper

ocean (top 700 m). Meanwhile, the ocean observing system has notably evolved with the implementation of the Argo floats network, thanks to which an unprecedented sampling of the subsurface temperature and salinity was achieved.

Regional ocean simulations or analysis systems need open boundary conditions at the domain boundaries, which may be provided by ocean reanalyses (e.g. Sotillo et al., 2015). Additionally, downstream applications (e.g. biogeochemical, fishery and larval dispersal models) require physical fields as input, preferably coming from an ocean reanalysis (e.g. Lazzari et al.,

2012; Meliá et al., 2013).

Improvements in ocean reanalyses are also required in the context of producing Earth system reanalyses for the atmosphere and ocean. "Strongly coupled" data assimilation methods are active research topics that will likely improve the use of observations for a variety of applications such as short- and long- term predictions (Laloyaux et al., 2015) and climate reanalyses (e.g. Dee et al., 2014). This implies that the optimal configuration of ocean retrospective analyses will benefit not

only ocean applications strictly speaking but also Earth system assimilative experiments.

All the applications listed above clearly motivate developments and production of ocean reanalyses with possibly short delay of dissemination beyond real time. The Fondazione CMCC (Centro Euro-Mediterraneo sui Cambiamenti Climatici) has devoted efforts in the last decade to build a state-of-the-art ocean physical reanalysis system. First reanalyses were produced at relatively coarse resolution (about 2 degrees) with the assimilation of hydrographic profiles only through an Optimal

Interpolation scheme (Bellucci et al., 2007; Masina et al., 2011). Since then, the data assimilation system has been upgraded to a three-dimensional variational scheme that allowed the assimilation of altimetry and remotely sensed data (Storto et al., 2011; Storto et al., 2013). The system has then been extensively validated and retuned, especially in the formulation and estimation of background-error covariances (Storto et al., 2014). Fostered by the European Union funded MyOcean project and its follow-ups (Masina et al., 2015), the resolution of the CMCC reanalysis system (called C-GLORS hereafter, i.e. the

CMCC Global Ocean Reanalysis System) has been increased to the eddy-permitting resolution (approximately 1/4 degree on the horizontal), as documented by Storto et al. (2016a) and has been used in a variety of different studies on the global ocean, ranging from semi-enclosed seas characterization (Cessi et al., 2014), to decadal studies on the Atlantic Ocean circulation and heat budget (Ezer, 2015; Yang et al., 2016).



C-GLORS was already released through the MyOcean portal (now at http://marine.copernicus.eu) since 2010, and upgraded along time. Aim of this paper is to describe and assess the improvements present in the latest released version of C-GLORS released (v5) and compare it with its predecessor. The article is structured as follows: Section2 describe the reanalysis system focussing on the changes with respect to C-GLORS v4. Section 3 presents selected results on the improvements

borne by the latest version. Section 4 summarizes and discusses the main achievements. The Appendix presents in details the reanalysis data product. Version 5 of C-GLORS will be updated in delayed time mode, typically with an approximate 6 months delay beyond present time due to the dissemination of quality-checked observational data.

## 2 C-GLORS reanalysis system

In this Section we briefly describe the reanalysis system, with the aim of highlighting the changes with respect to version 4.
A detailed description of the reanalysis system for v4 is available in Storto et al. (2016a). Storto et al. (2014) describes also the choices for background-error covariances in C-GLORS.

### 2.1 General Description

The reanalysis spans the period 1979-2014, although data are released from 1980 onwards as the first year is affected by an initial adjustment. The reanalysis system includes a i) three-dimensional variational data assimilation scheme called
OceanVar (see next Section) that assimilates hydrographic profiles from the U.K. MetOffice Hadley Center EN3 dataset (Ingleby and Huddleston, 2007) until 2012 and EN4 dataset (Good et al., 2013) afterwards, along-track altimetric observations provided by AVISO following Storto et al. (2011); ii) the NEMO ocean model (Madec, 2008), configured at about $1/4$ degree of resolution using a tripolar grid, with 50 vertical depth levels and partial steps (Barnier et al., 2006) and coupled to the LIM2 sea-ice model (Fichefet and Morales Maqueda, 1997) with elasto-visco-plastic (EVP) sea-ice rheology
(Bouillon et al., 2009); iii) a nudging scheme that assimilates space-borne sea-surface temperature observations supplied by NOAA (Reynolds et al., 2007) and sea-ice data; iv) a large-scale bias-correction (LSBC) scheme that corrects the model tendencies to limit the large scale biases induced by the NEMO model and the atmospheric forcing.
C-GLORS is forced by the ECMWF ERA-Interim atmospheric reanalysis (Dee et al., 2011) using the bulk formulas from Large and Yeager (2004). ERA-Interim provides three-hourly fields of temperature and humidity at 2 meters and wind at 10
meters and daily mean fields of shortwave and longwave radiation and total and solid precipitation. The shortwave radiation is modulated through the scheme of Bernie et al. (2007) to reproduce its diurnal cycle. Storto et al. (2016a) show the positive contributions of all assimilation components (OceanVar, surface nudging, LSBC) to the performance of C-GLORS. More details on the data assimilation formulation are provided in the next Section.

### 2.1.1 OcenVar data assimilation

The data assimilation system used in C-GLORS is a three-dimensional variational (3DVAR) scheme called OceanVar (Dobricic and Pinardi, 2008; Storto et al., 2014). The assimilation time-window is 7 days long and the frequency of the



assimilation steps is also 7 days. The 3DVAR scheme minimizes a cost function $J$ given with the incremental formulation (i.e. with minimization performed over $\delta x = x - x^b$):

$$J(\delta x) = \frac{1}{2}\delta x^T \boldsymbol{B}^{-1}\delta x + \frac{1}{2}(\mathbf{H}\delta x - \mathbf{d})^T \boldsymbol{R}^{-1}(\mathbf{H}\delta x - \mathbf{d}), \tag{1}$$

where $\boldsymbol{B}$ and $\boldsymbol{R}$ are the background- and observation- error covariance matrices, $\mathbf{d}$ is the vector of misfits calculated using the non-linear observation operator, $\mathbf{H}$ is the tangent-linear version of the observation operator. The analysis increment is found for $\delta x$ at the minimum of $J$. In order to avoid the inversion of $\boldsymbol{B}$ and precondition the minimization, the cost function is minimized over the space of the control vector $v$, with $\delta x = \boldsymbol{V}v$ and $\boldsymbol{B} = \boldsymbol{V}\boldsymbol{V}^T$, such that the cost function becomes

$$J(v) = \frac{1}{2}v^T v + \frac{1}{2}(\mathbf{H}\boldsymbol{V}v - \mathbf{d})^T \boldsymbol{R}^{-1}(\mathbf{H}\boldsymbol{V}v - \mathbf{d}). \tag{2}$$

The square-root background-error covariance matrix is decomposed in several operators:

$$\boldsymbol{V} = \boldsymbol{V}_\eta \boldsymbol{V}_H \boldsymbol{V}_V \tag{3}$$

where $\boldsymbol{V}_V$ is the vertical covariances operator, modeled through 10-mode season-dependent multivariate empirical orthogonal functions (EOFs) of temperature and salinity, namely $\boldsymbol{V}_v = \boldsymbol{U}\boldsymbol{\Lambda}^{1/2}$, with $\boldsymbol{U}$ the matrix containing eigenvectors and $\boldsymbol{\Lambda}$ the diagonal matrix containing eigenvalues. The horizontal operator $\boldsymbol{V}_H$ is modeled through a first-order recursive filter with inhomogeneous correlation length-scales (Storto et al., 2014) and $\boldsymbol{V}_\eta$ is the sea-level operator that analytically computes the sea level increment from increments of temperature and salinity using a local hydrostatic balances (Storto et al., 2011).

The observation error covariance matrix is assumed diagonal, i.e. observation errors are assumed independent between any pair of observations. For in-situ observations, the error accounts for instrumental accuracy and representativeness error and is defined as a function of the parameter (temperature or salinity), type of observation (bathythermographs, CTDs, floats or moorings), the depth and the location. For sea level anomaly, the error variance accounts for the satellite instrumental accuracy, the mean dynamic topography error (provided by CLS/AVISO) and the representativeness error. Usual climatology and background quality check are implemented in C-GLORS and described by Storto et al. (2016a).

## 2.2 Improvements with respect to the previous version

We review here the main improvements of C-GLORSv5 with respect to its predecessor C-GLORSv4. The upgrade of the system follows the validation exercise performed with C-GLORSv4, which highlighted the need of solving some deficiencies present in that version. The completion of the new version has also been fostered by the requirements of many users in term of augmented output frequency and variables. Below, we detail the changes for each of the reanalysis components.



### 2.2.1 Ocean and sea-ice model

The NEMO version used for C-GLORSv5 is 3.2. No major differences are implemented in this version of C-GLORS except for a tuning of the TKE (Turbulent Kinetic Energy) vertical mixing closure scheme. Following the work of Calvert and Siddorn (2013) and Megann et al. (2014), values for the background vertical viscosity and diffusivity were increased to
1.2E-4 and 1.2E-5 m2/s, respectively, from the original 1.0E-4 and 1.0E-5 m2/s, namely the vertical diffusivity and viscosity were increased by 20%. Additionally, the vertical decay of the TKE penetration was restored to the constant value of 10 m along the latitudes, opposed to the previous latitude-dependent formulation, found to deteriorate cold biases and warm biases at mid- and low- latitudes, respectively.

Additionally, C-GLORSv5 does not include any longer the ERA-Interim forcing correction described in Storto et al. (2012)
and Storto et al. (2016a). This choice reflects the need of avoiding specific post-processing of the atmospheric forcing required by near real-time production, along with the idea of preferring the use of atmospheric parameters in balance with each other rather than introduce imbalances due to the correction of radiative fluxes only.

### 2.2.2 Data Assimilation

The variational data assimilation system in C-GLORSv5 has been slightly modified from its predecessor reanalyses and in
particular both observational and background error covariances were retuned, plus a few additional improvements and bug fixes.

The tuning of in-situ observational errors was achieved by using data assimilation output statistics from C-GLORSv4 and applying the posterior method of Desroziers et al. (2006). Such a method relies on the cross-covariance statistics of observation misfits and assimilation residuals in observation space to diagnose background- and observation- errors in
observation space. It is broadly used in meteorological and oceanographic applications because of its simplicity, although limits of the methods are found in case of incorrectly prescribed spatial correlations or asynchronous tuning of background- and observation error variances (Menard et al., 2016).

Figure 1 shows the profiles of observational errors for in-situ observations before and after the tuning, for the four in-situ observation types assimilated in the system. While the shapes of the vertical profiles are very close in C-GLORSv4 and v5,
salinity errors were approximately halved in the upper ocean for all observation types, implying that salinity analysis increments in C-GLORSv4 were under-estimated, based on this posterior diagnostics. Differences between the two temperature error profiles depend on the type: while mooring (buoys) errors are reduced, Argo and CTD errors are increased and XBTs almost unchanged. The tuning in practice flattens the differences in errors between the observation types, leading to a closer profile for the four observing networks than in C-GLORSv4. This result can be interpreted with the dominant
effect of representativeness error with respect to the instrumental error in ocean observations, the former being reasonably more similar between different observing networks.



Background-error vertical covariances have also been tuned in C-GLORSv5. The modification consisted in introducing a corrective coefficient, by means of which covariances were modulated in time and space. The new vertical EOFs operator $\boldsymbol{V_v^*}$ is formulated as

$$\boldsymbol{V_v^*} = \alpha(t)\beta(\phi,\lambda)\,\boldsymbol{V_v} = \alpha(t)\beta(\phi,\lambda)\,\boldsymbol{U\Lambda^{1/2}} \tag{4}$$

where $\boldsymbol{V_v}$ is the uncorrected operator, as in C-GLORSv4, and $\alpha$ and $\beta$ are corrective coefficients, as a function of time or longitude and latitude, respectively. The specification of both the coefficients was performed by applying the Desroziers' method on the assimilation output statistics from C-GLORSv4, similarly to the tuning of the observational errors.

Figure 2 shows the new time-averaged background-error covariances after the application of the corrective coefficients for the temperature in the top 10 m of depth (top panel) and the difference with the corresponding standard deviations in C-
GLORSv4 (bottom panel). The use of the corrective factor generally reduces the covariances at global scale, with a particularly large reduction in the Tropical Pacific Ocean and within the North Pacific gyre. As discussed in Storto et al. (2014) and Storto et al. (2016a), the overestimation in the Tropical Pacific was a known issue, attributable to the dataset from which EOFs are calculated, which is formed by monthly mean anomalies. Indeed, such an approach by construction confuses the ocean variability with the ocean error variance. These two were found to diverge especially in the Tropical
oceans, where errors were over-estimated. Differently, some regions dominated by strong mesoscale activity (Gulf Stream, Kuroshio, Antarctic Circumpolar Current) exhibit an increase of background-error variances, suggesting that the method of using monthly mean anomalies may underestimate error variances in these areas due to the coarse temporal resolution of the monthly anomalies with respect to the weekly assimilation frequency.

Figure 3 shows the temporal coefficient $\alpha$, which is assumed to be globally uniform. Even though its global definition cannot
capture patterns associated to the geographical sampling of the observations, it mainly reproduces the change of the in-situ observing system, with a gradual background-error reduction during the deployment period of the Argo floats (2000s). $\alpha$ should be considered as a preliminary attempt to introduce flow-dependent information in the background-error covariance characterization of C-GLORS. At the end of the reanalysis period, the background-error standard deviation are reduced by about 35% with respect to their nominal value at the beginning of the reanalysis.

C-GLORS implements a background quality check procedure that rejects observations whose square departure exceeds a certain number of times the sum of the observational and background error variances. For the $i$-th observation, the observation retention criterion reads:

$$\left(y_i - H_i(x^b)\right)^2 \leq \alpha\left[\sigma_{o,i}^2 + \sigma_{b,i}^2\right] \tag{5}$$

with $\sigma_{o,i}^2$ and $\sigma_{b,i}^2$ the observation and background error variances, in observation space, and $\alpha$ the quality check threshold.
This scheme was also retuned, the threshold being increased from 6 to 9 to allow a greater number of assimilated observations, particularly important in areas of strong mesoscale variability.

The rejection of observations near the cost has also been softened, with a reduction of the minimum distance from shorelines from 75 to 15 km. Finally, a bug preventing the assimilation of observations located at a depth shallower than the first model





level (around 0.5 m) was fixed in C-GLORSv5, allowing a greater number of sea surface observations from in-situ profiles to be ingested. This change produced a 9% increase of assimilated observations.

### 2.2.3 Observational dataset

As C-GLORSv5 is conceived to be continuously updated in delayed time mode, a change of the observational data was made during the production of the reanalysis. In particular, from 1979 to 2012 the in-situ data were extracted from EN3 (Ingleby and Huddleston, 2007). From 2013, the reanalysis assimilates hydrographic profiles from EN4 (Good et al., 2013). A similar change concerns also altimetry data: two versions of the delayed time along-track dataset from CLS/AVISO with the upgrade occurring at the beginning of 2013. The MDT in C-GLORS is computed as long-term mean SSH (Storto e al., 2016a) from a twin experiment with the assimilation of in-situ profiles only. Because the two versions of assimilated AVISO SLAs are referenced to 1993-1999 and 1993-2012, respectively, at the beginning of 2013 the MDT was recalculated as 1993-2012 long-term mean from the previous 20 years of reanalysis and substituted to the 1993-1999 referenced MDT. This approach preserves the continuity of the MDT used and prevents from possible adjustments from 2013.

### 2.2.4 Large-scale bias correction

The large-scale bias correction (LSBC) scheme of C-GLORS relaxes the temperature and salinity large-scales towards large-scale uni-variate objective analyses of temperature and salinity measurements from hydrographic profiles. This scheme has proven successful for bias- and drift- correcting long-term diagnostics in C-GLORS, mostly arising from systematic errors in the forcing (Storto et al., 2016b). The new version of C-GLORS has a decreased temporal scale for use in this scheme, passing from 3 to 36 months. Preliminary tests showed that a 3-month time-scale was partly preventing the inter-annual variability of the heat content to evolve independently from the objective analyses, resulting in a too close reproduction of the latter. The new time-scale shows robust in preventing large-scale biases without jeopardizing the reanalysis inter-annual variability. This is shown in Figure 4, which reproduces the global ocean heat content in non-assimilative experiments covering the period 1958-1989 for different time scales of the large-scale bias correction. Without LSBC, the model drifts away from the initial conditions, due to weakly negative air-sea heat fluxes (Storto et al., 2016b). With a 3-month period, the variability of the modelled ocean heat content resamples very well that of EN4, while the 3-year time-scale prevents the simulation to drift without following closely EN4.

### 2.2.5 Sea-ice data assimilation

In C-GLORSv4, we assimilated sea-ice concentration analyses retrieved from the DMSP constellation of passive microwave radiometers, provided by NSIDC through the use of the NASA Team algorithm (Cavalieri et al., 1999). While C-GLORSv4 has a good representation of seasonal and inter-annual variations in sea-ice extent and area, the ice volume was characterized by anomalous variability, which we found caused by the assimilation of sea-ice concentration in highly ice-covered areas leading to spurious increases in ice thickness. This is a common problem when sea-ice concentration is assimilated but no



constraint is applied to sea-ice thickness (e.g. Tietsche et al., 2013). In C-GLORSv5 we introduce a nudging scheme in order to weakly constrain sea-ice thickness in the Arctic Ocean. Model sea-ice thickness is relaxed towards PIOMAS data (Zhang and Rothrock, 2003) with a 15-day relaxation time scale. Although PIOMAS is a reanalysis itself that assimilates only ice concentration data, it has been extensively validated and proves a reliable tool for the reconstruction of the Arctic sea-ice

volume in the past decades. This approach proves able to mitigate the spurious variability of sea-ice volume, although future work should be done to implement robust multi-variate corrections of sea-ice parameters in polar regions.

### 2.2.6 Initialization

The initialization strategy in C-CGLORSv5 was changed. While C-GLORSv4 was initialized through a spinup with repeated atmospheric forcing relative to 1978 for ten years, C-GLORSv5 was initialized using the 1979-1982 mean January

conditions. This led to a weaker shock at the beginning of the reanalysis, allowing the dissemination of the reanalysis data from 1980.

### 3 Selected results

In this Section, we review the results that summarize the main improvements of C-GLORSv5 with its predecessor (C-

GLORSv4).

### 3.1 Verification skill scores

Verification of model fields against Argo floats from EN4 was conducted for both C-GLORSv4 and C-GLORSv5. Outputs fields are compared to the data before the assimilation, namely the floats used represent a fairly independent dataset. We report in Figure 5 vertical profiles of bias and RMSE for salinity and temperature. The validation results suggest that

differences between the two versions are not large. In particular, C-GLORSv5 is able to reduce the warm temperature bias in the first 100 m of depth, which leads to a corresponding RMSE decrease. The improvement of the upper ocean temperature is particularly evident in the tropical region (not shown). On the other hand, salinity skill scores are characterized by a saline bias in the upper part of the ocean, which slightly increase in C-GLORSv5.

Figure 6 shows the sea surface temperature RMSE difference between C-GLORSv4 and v5. C-GLORSv5 improves the

representation of the SST especially at low latitudes and in particular in the Pacific Ocean. The latest version appears however less skilful around the Gulf Stream and Kuroshio Extension, the former probably due to a misplacement of the main current system. Less meaningful the slight worsening at very high latitudes, where skill score statistics may be affected by different sea-ice covered areas. The global improvement on the SST in terms of RMSE decrease is 3.2%. It peaks to 7.7% in the Tropics (from 30S to 30N), where the C-GLORSv4 warm bias in the Pacific Ocean section is reduced in C-GLORSv5

(not shown).





### 3.2 Global ocean heat content and steric sea level rise

Among the main climate change signatures on the oceans, the ocean heat content has increased notably during the last decades (Levitus et al., 2012). Related to it, the change in the global steric sea level represents a fundamental diagnostics that ocean reanalyses need to capture, whereas the global change in the halosteric component is rather neutral at global scale (Stammer et al., 2013). Figure 7 shows the global steric sea level change (calculated over the top 2000 m, for consistency with observational datasets) in C-GLORSv4, C-GLORSv5, the NOAA/NODC in-situ based estimates (Levitus et al., 2012), and the dataset proposed by Storto et al. (2015) for the period 2003-2011 merging satellite altimetry and gravimetry data. For the latter, global steric sea level is inferred by the difference of global sea level change from altimetric missions minus global barystatic sea level change from the GRACE gravimetric mission. C-GLORV5 data start in 1980 unlike C-GLORSv4 (1982). The 1980-1982 years are characterized by an abrupt steric sea level decrease following El Chichon volcanic eruption (Church et al., 2005). Similarly, a decrease at the beginning of the 1990s follows the Pinatubo eruption, whose effects appear to last longer in C-GLORSv5. A significant difference between the two versions of C-GLORS occurs in the last decade (from 2005 onwards), where C-GLORSv5 reduces the warming hiatus (Trenberth and Fasullo, 2013) present in C-GLORSv4. This feature leads C-GLORSv5 to have a 2003-2011 trend higher than C-GLORSv4 (0.57 vs 0.02 mm/yr) and much closer to the satellite independent estimates (0.77 mm/yr), with an increased temporal correlation with the latter as well.

To further deepen the main changes of C-GLORSv5 with respect to v4 in terms of the last decade heat content increase, Figure 8 shows the 2005-2013 top 2000m heat content trend in C-GLORSv5, in the NOAA/NODC datasets and trend differences between C-GLORSv5 and v4. From this sketch, it appears that heat content trend is in close agreement with the NOAA/NODC estimates. In particular, the increase of linear trends in C-GLORSv5 with respect to its predecessor occurs mostly in the North Atlantic and, only partly, in the Tropical Pacific Ocean. The heat content difference in the second half of 2000s between the two reanalysis versions is mostly contributed by the North Atlantic Ocean, whose trends now closely resembles those estimated by the NOAA/NODC; this improvement is likely due to the weakening of the bias correction scheme in association with the decrease of background-error covariances in the last decade (Figure 3), which jointly strengthen the weight given to temperature measurements from 2000 onwards.

### 3.3 Atlantic meridional overturning circulation and associated heat transport

We compare in this section the representation of the Atlantic Ocean meridional overturning circulation (AMOC) estimated from C-GLORSv4 and v5 with the RAPID-MOC array estimates (Rayner et al., 2011). This is shown in the top panel of Figure 9 in terms of the maximum of the meridional stream-function at 26N. During the period of availability of the RAPID array data (2004-2013), C-GLORS v5 has a slightly better correlation with the observational estimates with respect to C-GLORSv5 (0.83 vs 0.80 for the monthly means and 0.87 vs 0.82 for the yearly means).



More importantly, the long-term average of C-GLORSv5 is 1.3 Sv higher than in C-GLORSv4, getting closer to the observational estimates and reducing the under-estimation of AMOC in the previous reanalysis, due to an under-estimation of the Florida Current and the interior southward transports (Storto et al., 2016a). In particular, the largest effect of the new reanalysis configuration involves the strengthening of the southward transport of the upper North Atlantic Deep Waters at

26N (NADW, defined according to McCarthy et al., 2015), which increases from -14.6 to -16.0 Sv. The increase of AMOC in C-GLORSv5 is more evident up to the beginning of 2000s; afterwards, the two versions show very similar values, i.e. the mean increase of AMOC is significant in data-poor periods, suggesting that is likely due to refinement of large-scale bias correction, atmospheric forcing, model vertical physics or sea-ice treatment rather than the tuning of the 3DVAR scheme.

As a consequence, the meridional heat transport (MHT) at 26N appears slightly larger in C-GLORSv5 (1.0 vs 0.9 PW)

compared to C-GLORSv4, closer to the mean value derived from the RAPID-MOCHA heat flux array (1.23 PW, Johns et al., 2011). The bottom panel of Figure 9 reproduces the timeseries of MHT for the period 1982-2013. Note however that the under-estimation of the heat transport in the reanalysis compared to RAPID estimates might arise from the geostrophic approximations implicit in the RAPID calculations (Stepanov et al., 2016).

### 3.4 Sea-ice reconstruction

Sea-ice areas and extents from C-GLORSv5 are shown in Figure 10 for both the Antarctic and Arctic oceans and compared to C-GLORSv4. Sea-ice areas between version 4 and 5 are in close agreement for both polar regions, as a consequence of the same sea-ice concentration data assimilation scheme and observational dataset. On the contrary, the sea-ice volume variability and trend significantly change between the two versions, especially in the Arctic region.

For the latter, the spurious variability present in version 4 – sea-ice volume minima during 1987-1988 and maximum during 2006 – is replaced by a realistic and smaller inter-annual variability, with minima correctly occurring during September 2007 and 2012. This obviously improves the estimates of the Autumn (October/November) sea-ice volume trends, equal to -1190 (-295) $Km^3 yr^{-1}$ for the 1980-2014 (2004-2008) period, opposed to C-GLORSv4 that shows -4025 (-114) $Km^3 yr^{-1}$, in close agreement with estimates from PIOMAS for the entire period (-280 $Km^3 yr^{-1}$) or from ICESat for the 2004-2008 period (-

1240 $Km^3 yr^{-1}$). The weak constraint on the sea-ice thickness thus proves robust to fix the spurious variability problems and provides sea-ice volume estimates in agreement with observation-based products.

Similarly, sea-ice volume spurious variability, albeit smaller, is recovered in the Antarctic region as well. This leads to better sea-ice volume trend estimates, equal to +34 $Km^3 yr^{-1}$ for the entire period and -250 $Km^3 yr^{-1}$ for the spring periods 2003-2008, in closer agreement to other data assimilative experiments (+36 $Km^3 yr^{-1}$ during 1980-2008, Massonet et al., 2013)

and ICESat derived estimates (-266 $Km^3 yr^{-1}$ across 2003-2008 springs, Kurtz and Markus, 2012) with respect to C-GLORSv4 (+77 and -698 $Km^3 yr^{-1}$, respectively), where an over-estimations of trends in both periods occurred. It should be noted that here no sea-ice thickness constraint is applied because of lack of thickness data in the Antarctic region, suggesting



that the re-tuning of the data assimilation system is here responsible for the improved interannual variability of the sea-ice volume.

Based on this assessment, C-GLORSv5 proves a reliable tool for investigating the ocean and sea-ice interannual variability in polar regions.

### 3.5 Air-Sea heat flux

A possible application of ocean reanalyses is the study of heat budget (e.g. Yang et al., 2016). Therefore, validation of air-sea heat fluxes represents a necessary step for this kind of applications. C-GLORSv4 was affected by an under-estimation of air-sea net heat flux, due, among other factors, to the radiative fluxes corrections that were unbalanced with turbulent atmospheric forcing, inducing spuriously negative net heat fluxes at low- and mid- latitudes.

The use of uncorrected forcing along with the tuning of the data assimilation system leads to a reduced surface bias in C-GLORSv5, which translates to better air-sea heat fluxes. This feature is shown in Figure 11, in terms of zonally averaged monthly climatology of net heat flux for C-GLORSv5, and differences between C-GLORSv4 and OAFlux (Yu and Weller, 2007), and C-GLORSv5 and OAFlux. OAFlux is a satellite based air-sea fluxes dataset that relies on the ISCCP (Zhang et al., 2007) for the net radiative fluxes at the sea surface. Although the large uncertainty of net heat flux from observational estimates and reanalyses (Valdivieso et al., 2015), the use of this independent dataset for the reanalysis validation suggests that C-GLORSv5 has the net heat flux in the Tropics (year-round) in closer agreement to OAFlux with respect to C-GLORSv4. Boreal winter fluxes in the Southern Ocean, under-estimated in C-GLORSv4, also appear now in close agreement.

Improved heat fluxes estimates emerge also in the global net heat flux average. While the previous C-GLORS reanalysis system was characterized by a large negative imbalance (Valdivieso et al., 2015) equal to -11.2 W/m2, C-GLORS exhibits a much smaller imbalance of -3.9 W/m2. Turbulent (latent and sensible) heat fluxes are in practice unchanged between the two versions, the differences of which arise from the radiative fluxes, gaining in particular 2.7 W/m2 from the net longwave flux, and 4.9 W/m2 from the net shortwave flux. Both increases can be ascribed to the turning off of the radiative forcing correction; for the longwave flux increase, the reduction of the warm tropical bias proves also to play a role.

### 4 Summary and conclusions

We describe in this article the C-GLORSv5 global ocean reanalyses dataset, available from www.pangaea.de (https://dx.doi.org/10.1594/PANGAEA.857995). The dataset covers the 1980-2014 period and is being updated in delayed time (with approximately two years delay, due to the delayed time reprocessing of all input observational datasets). As a global ocean product, C-GLORSv5 may be useful for a large variety of applications that range from climate monitoring to ocean- and process- related studies, to downstream applications (biogeochemistry, fisheries, sediment, dispersion, etc,) and regional downscaling, to initialization of long-range prediction systems, etc..



Following the lessons learned during the validation exercise of the previous version, this version of C-GLORS contains a number of significant improvements, among which a complete retuning of the data assimilation system (especially background- and observation- errors), retuned schemes for the large-scale bias correction and sea-ice data assimilation, as well as an updated version of input datasets and an improved initialization strategy. Additionally, a background-error

temporal inflation coefficient has been introduced to mimic the reanalysis accuracy increase corresponding to the Argo floats deployment. The C-GLORSv5 reanalysis system also produced daily mean outputs available on-demand, further to standard monthly mean outputs detailed in the Appendix.

While it is difficult to disentangle the individual effect of the reanalysis upgrades on the resulting reanalysis accuracy, we have focussed the validation exercise presented in Section 3 on highlighting the main changes and improvements of C-

GLORSv5 with respect to its predecessor C-GLORSv4. Validation skill scores appear comparable between the two products, although C-GLORSv5 presents an improved representation of the upper ocean temperature mean state and variability, particularly evident in the Tropical region and leading to an improved representation of the air-sea heat fluxes.

The heat content change in the last decade also appears in closer agreement with independent estimates, likely due to the combined effect of the improved surface state and re-tuning of assimilation and bias-correction components of C-GLORSv5.

This feature appears evident in the North Atlantic Ocean. As a consequence, the representation of the Atlantic Ocean overturning circulation is also concerned, with C-GLORSv5 showing a strengthening which brings it closer to the RAPID array estimates.

As no constraint was applied to sea-ice thickness in C-GLORSv4, the absence of sea-ice conservation methods – as the ones proposed for instance by Tietsche et al. (2013) – caused drifts in the sea-ice volume timeseries. An intermediate solution

consisting in a weakly relaxation towards realistic sea-ice thickness data was implemented in C-GLORS for the Arctic region, which led to inter-annual variability for sea-ice volume in accordance to many model and observational studies. In future releases of C-GLORS, multi-variate sea-ice assimilation will be considered, as it emerges as the desired strategy to simultaneously correct sea-ice parameters from sea-ice concentration data with multivariate balances, without the recourse to empirical nudging schemes for the surface state assimilation.

Overall, the reanalysis proves a valuable tool not only for regional down-scaling and down-streaming applications, but also for process- and budget- oriented studies concerning for instance heat exchanges, air-sea-fluxes and sea-ice energy. Additionally, climate-oriented ocean simulations (e.g. the ocean components of the Coupled Model Intercomparison Project, CMIP, e.g. Taylor et al., 2012), may benefit from this dataset for comparison and validation purposes.

**Appendix: Product Description**

The following Table 1 describes the ocean and sea-ice parameters available from C-GLORSv5 at doi:**10.1594/PANGAEA.857995** (https://doi.pangaea.de/10.1594/PANGAEA.857995). All variables are available as monthly means over the period 1980-2014 in NetCDF format. 2D variables are interpolated onto a global regular grid at 1/2



degree of horizontal resolution. Due to their large size, three-dimensional variables (temperature, salinity and currents), high-frequency variables (daily means) or high-resolution variables (native 1/4 degree grid) are available on-demand through requesting them at www.cmcc.it/c-glors/index.php?sec=8.

5     **Table 1. C-GLORS5 disseminated ocean and sea-ice parameters.**

| 2D Variables | | |
|---|---|---|
| **Variable Name** | **Long Name/Meaning** | **Units** |
| barotropic_streamfunction | Barotropic Streamfunction | m3 s-1 |
| freshwater_content_0-300 | Freshwater Content in the top 300 m | m |
| freshwater_content_0-700 | Freshwater Content in the top 700 m | m |
| freshwater_content_0-2000 | Freshwater Content in the top 2000 m | m |
| freshwater_content_0-bottom | Total Freshwater Content | m |
| heat_content_0-300 | Heat Content in the top 300 m | J m-2 |
| heat_content_0-700 | Heat Content in the top 700 m | J m-2 |
| heat_content_0-2000 | Heat Content in the top 2000 m | J m-2 |
| heat_content_0-bottom | Total Heat Content | J m-2 |
| latent_heatflux | Latent Heat Flux | W m-2 |
| meridional_wind_stress | Meridional Wind Stress | N m-2 |
| mixed_layer_depth_d010 | Mixed Layer Depth (0.10 Kg m-3 density criterion) | m |
| net_longwave_heatflux_downwards | Net Longwave Heat Flux (downwards) | W m-2 |
| net_shortwave_heatflux_downwards | Net Shortwave Heat Flux (downwards) | W m-2 |
| net_surface_heatflux_downwards | Net Heat Flux (downwards) | W m-2 |
| net_surface_waterflux_upwards | Net Water Flux (downwards) | Kg m-2 s-1 |
| seaice_concentration | Sea Ice Concentration | 0-1 |
| seaice_meridional_velocity | Sea Ice Meridional Velocity | m s-1 |
| seaice_thickness | Sea Ice Thickness | m |
| seaice_zonal_velocity | Sea Ice Zonal Velocity | m s-1 |
| sea_surface_height | Sea surface height | m |
| sea_surface_meridional_current | Sea Surface Meridional Current | m s-1 |
| sea_surface_salinity | Sea Surface Salinity | psu |
| sea_surface_temperature | Sea Surface Temperature | degC |
| Sea_surface_zonal_current | Sea Surface Zonal Current | m s-1 |
| sensible_heatflux | Sensible Heat Flux | W m-2 |





| zonal_wind_stress | | Zonal Wind Stress | N m-2 |
|---|---|---|---|
| land_sea_mask | | Land Sea Mask (1 over sea) | 0-1 |
| **Timeseries and integrated variables** | | | |
| **Variable Name** | **Dimensions** | **Long Name/Meaning** | **Units** |
| ftransp | Time | Freshwater transport across selected sections* | Sv |
| htransp | Time | Heat transport across selected sections* | PetaWatt |
| Vtransp | Time | Volume transport across selected sections* | Sv |
| icearea_antarctic | Time | Sea Ice Area in the Antarctic region (net area of sea-ice where concentration is greater than 15%) | m2 |
| icearea_arctic | Time | Sea Ice Area in the Arctic region (net area of sea-ice where concentration is greater than 15%) | m2 |
| iceext_antarctic | Time | Sea Ice Extension in the Antarctic region (extension of sea-ice where concentration is greater than 15%) | m2 |
| iceext_arctic | Time | Sea Ice Extension in the Arctic region (extension of sea-ice where concentration is greater than 15%) | m2 |
| icevolume_antarctic | Time | Sea Ice Volume in the Antarctic region | m3 |
| icevolume_arctic | Time | Sea Ice Volume in the Arctic region | m3 |
| amoc | Latitude, Depth, Time | Atlantic Ocean Meridional Streamfunction | Sv |
| amoc_26N | Depth, Time | AMOC at 26N | |
| max_amoc_26N | Time | Maximum of the AMOC at 26N | Sv |
| mht_a | Latitude, Time | Meridional Heat Transport in the Atlantic Ocean | PetaWatt |
| mht_g | Latitude, Time | Global Meridional Heat Transport | PetaWatt |
| amoc | Latitude, Depth, Time | Atlantic Ocean Meridional Streamfunction | Sv |
| halosteric | Time | Global Halosteric sea level anomaly | m |
| thermosteric | Time | Global Thermosteric sea level anomaly | m |
| steric | Time | Global Steric sea level anomaly | m |

* The selected sections are: Antarctic Circumpolar Current in correspondence of Australia at 143W (ACC_AUS); Antarctic Circumpolar Current in correspondence of Drake Passage at 68E (ACC_DRK); Antarctic Circumpolar Current in correspondence of South Africa at 30W (ACC_SAF); Bering Strait, 68N (BER); Greenland Sea at 60N (GIN); Labrador Sea at 60N (LAB); Indian Ocean at 32S (IND); Indonesian Throughflow at 102W-114W (ITF); North Atlantic Ocean at 35N (NAT); North Pacific Ocean at 24N (NPA); South Atlantic Ocean at 11S (SAT); South Pacific Ocean at 32S (SPA).



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



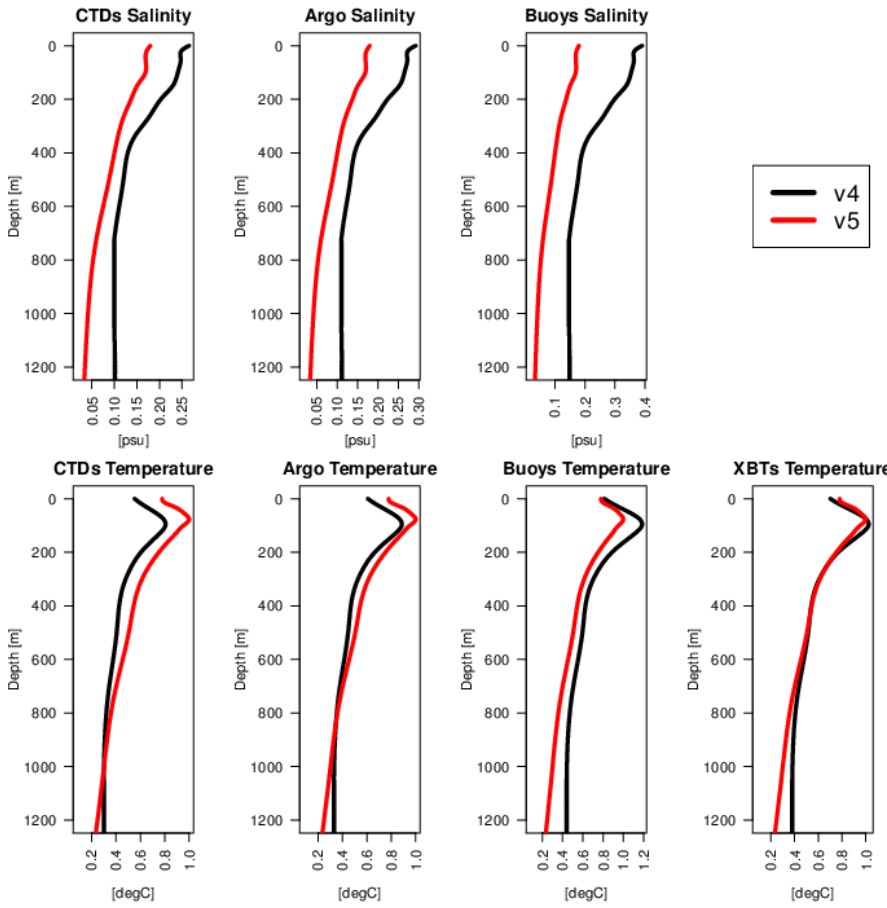

**Figure 1. Observation error profiles for use in the in-situ data assimilation in C-GLORS v4 (black lines) and v5 (red lines), as a function of parameter and observation type.**



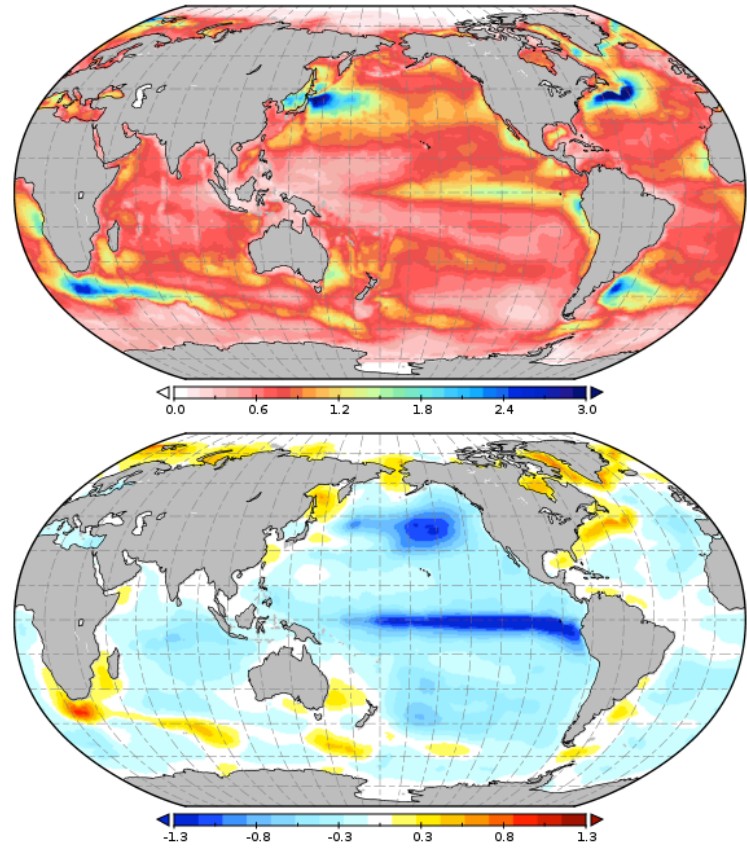

**Figure 2. Season-averaged temperature background-error standard deviation in the top 10 m of depth in C-GLORSv5 (top panel) and difference with respect to the background errors used in C-GLORSv4 (bottom panel).**



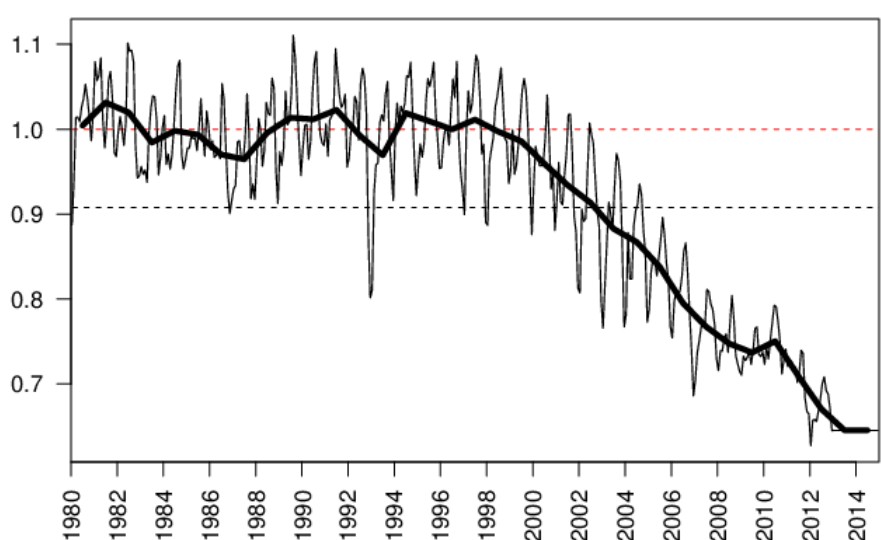

**Figure 3. Background error standard deviation temporal inflation (coefficient $\alpha(t)$ in Equation (4)). Thin (thick)**

5    **lines refer to monthly (yearly) means. Black dashed line corresponds to the long-term mean of $\alpha(t)$, while red dashed**

**line to the value of 1, i.e. the reference value before the application of the inflation.**





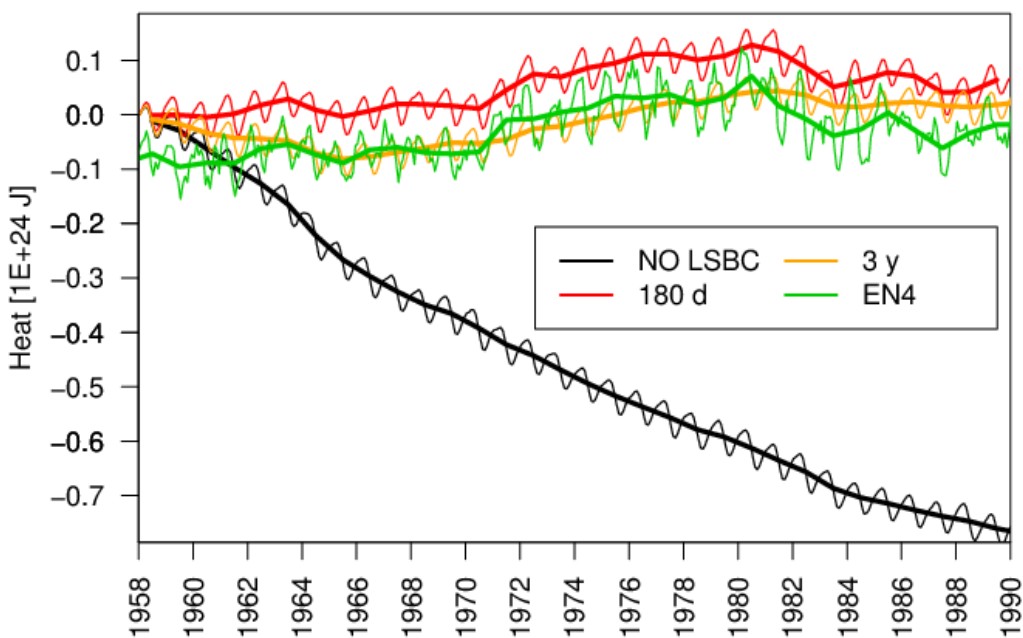

5  Figure 4. Total ocean heat content in a set of preliminary experiments during 1958-1990 performed to test the impact
of the large-scale bias correction (LSBC) time-scale. Thin (thick) lines refer to monthly (yearly) means. Black lines
refer to the experiment without LSBC; red and orange lines to the experiment with LSBC and a time-scale of 3
months and 3 years, respectively. Also shown the heat content computed from the UK MetOffice EN4 dataset (Good
10  et al., 2013)




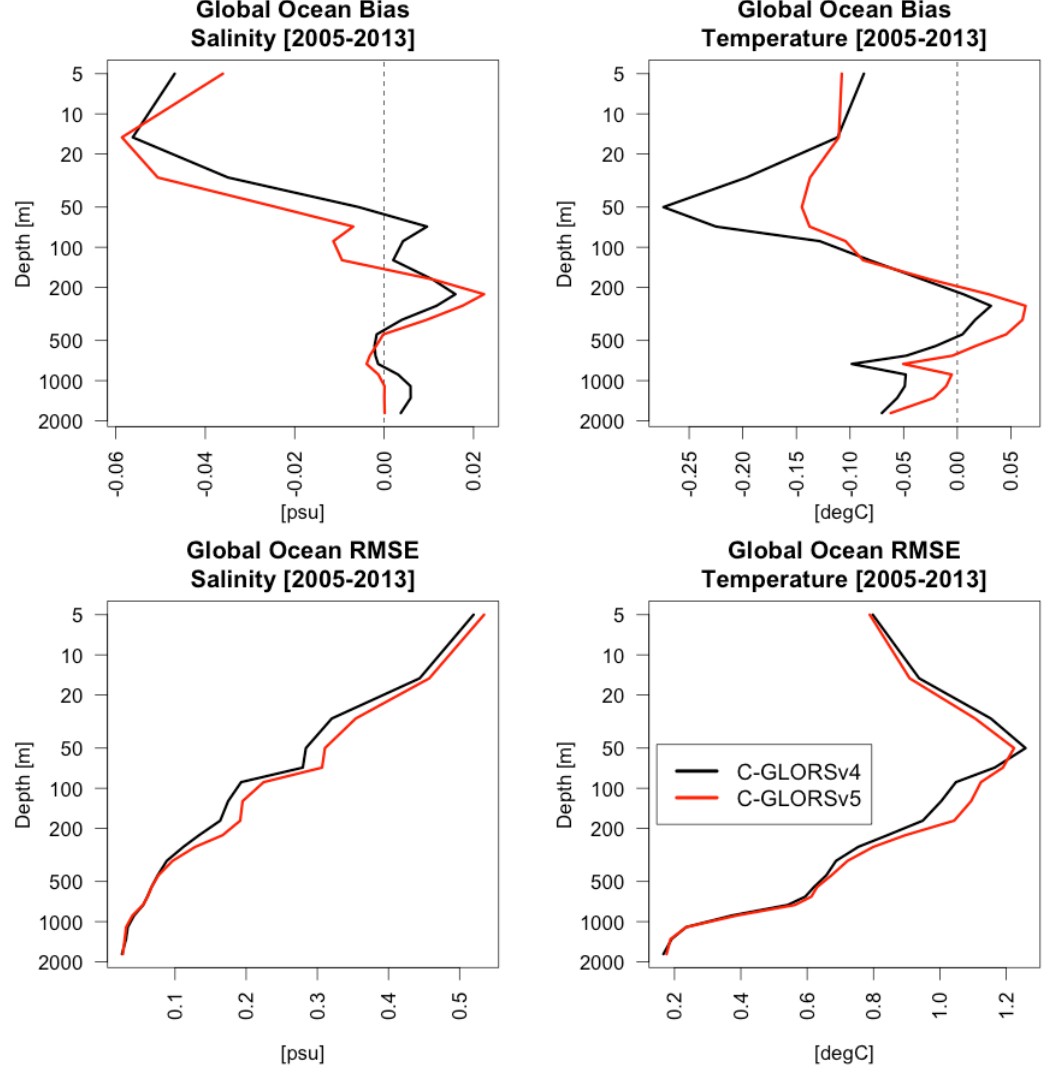

**Figure 5.** Verification skill scores (bias and RMSE) of C-GLORSv4 and C-GLORSv5 against Argo floats from EN4 over the period 2005-2013 as a function of ocean depth. Units are psu and degrees Celsius for salinity and temperature, respectively. Bias is defined as observation minus model fields.





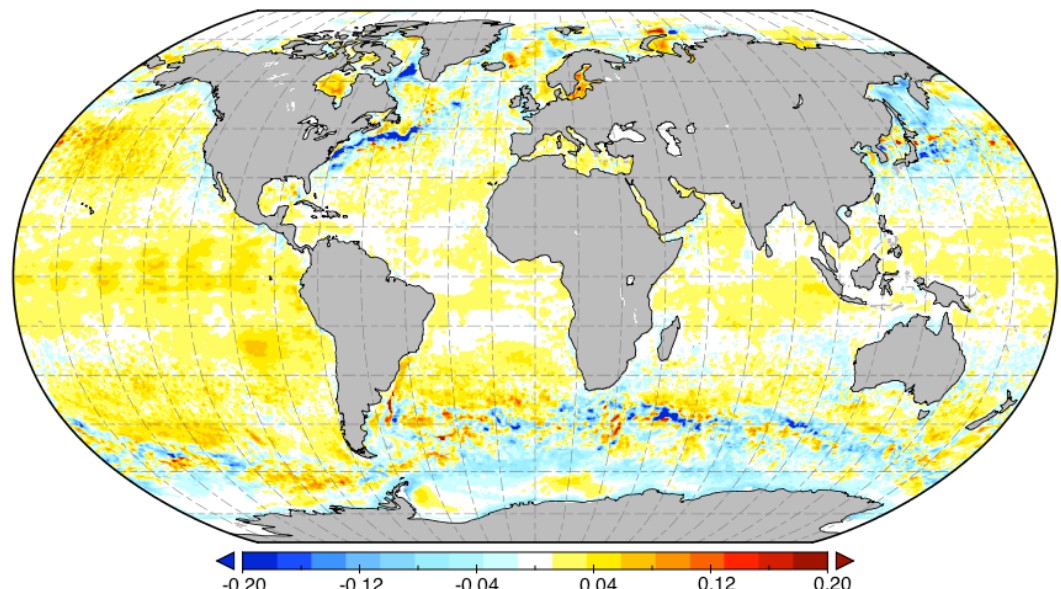

**Figure 6. SST Root Mean Square error difference of C-GLORSv4 minus C-GLORSv5, against NOAA SST ¼ daily
analyses (Reynolds et al., 2007). Positive (negative) differences indicate that C-GLORSv5 outperforms
(underperforms) C-GLORSv4. Units are degrees Celsius.**





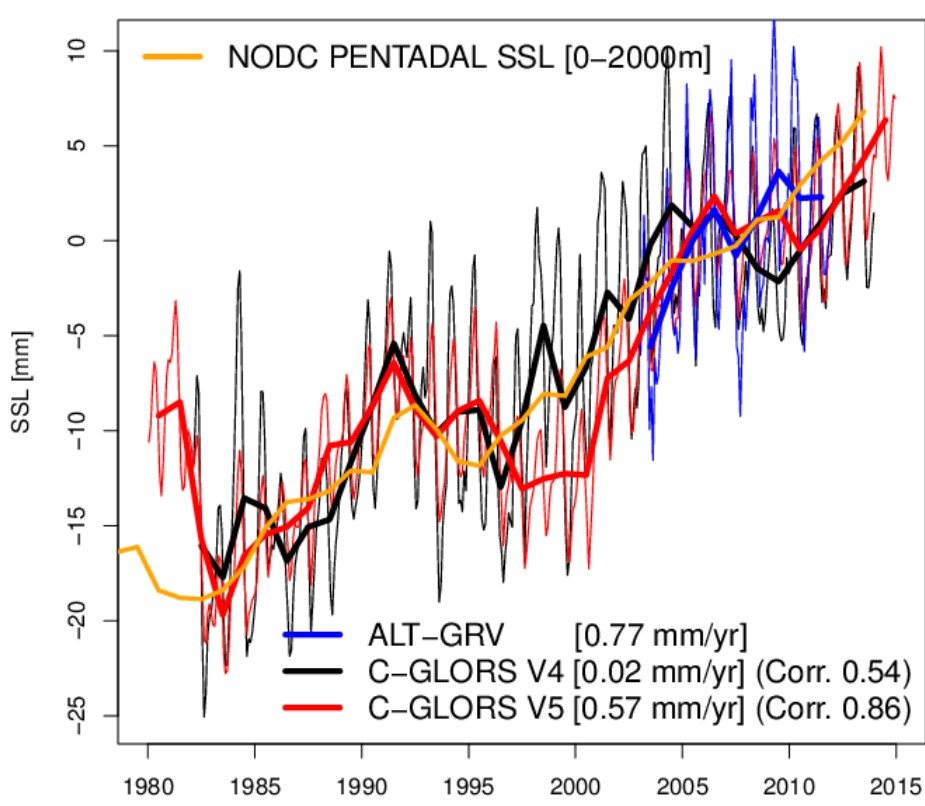

**Figure 7.** Global ocean steric sea level (calculated over the top 2000m) for the reanalyses C-GLORSv4 and v5, along with estimates from satellite altimetry and gravimetry (ALT-GRV, Storto et al. 2015) and NODC pentadal estimates (Levitus et al., 2012). The legend reports linear trends for the period 2003-2011 from the reanalyses and ALT-GRV, and the temporal correlation of the reanalyses with respect to ALT-GRV.





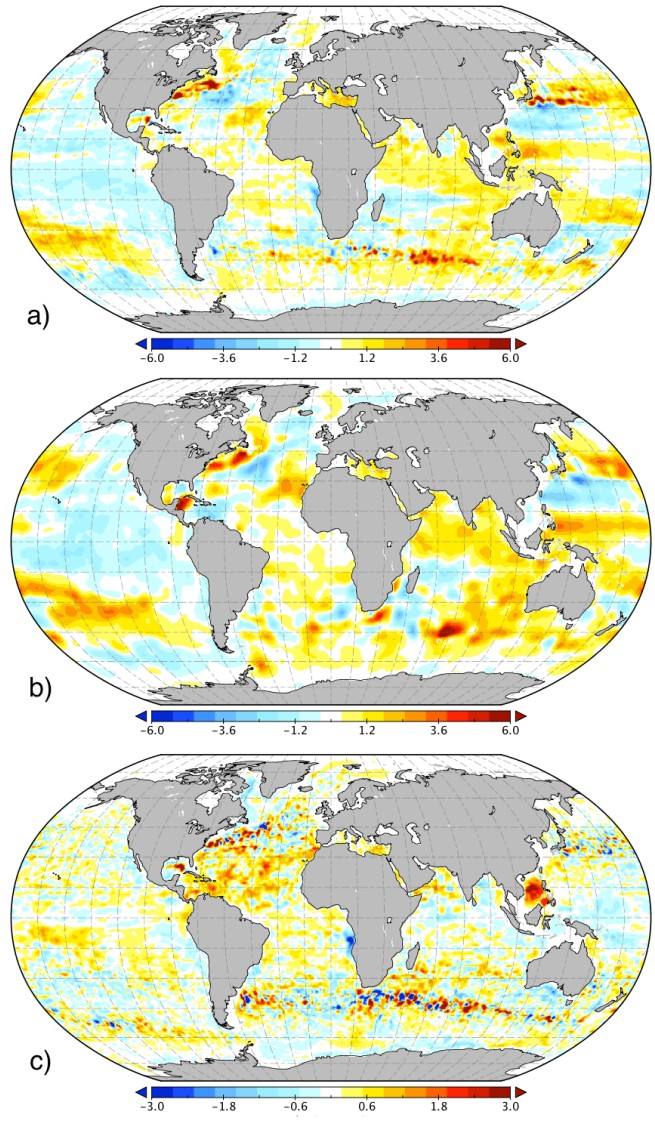

**Figure 8.** Top 2000 m heat content linear trend over the period 2005-2013 in C-GLORSv5 (panel a), NOAA/NODC data from Levitus et al. (2012) (panel b) and difference between C-GLORSv5 and C-GLORSv4 (panel c). Units are 1.E+18 Joules/year.



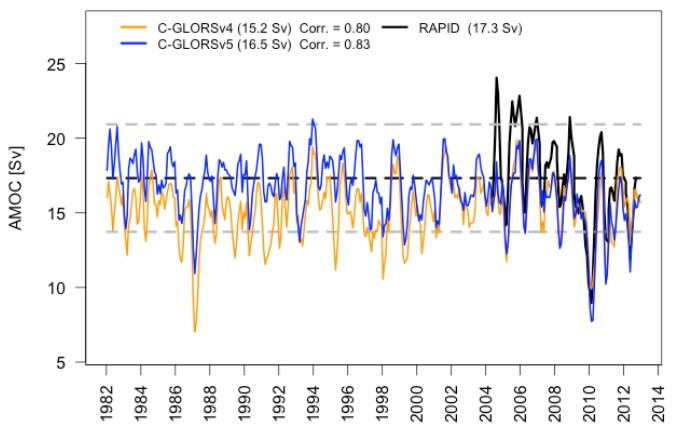

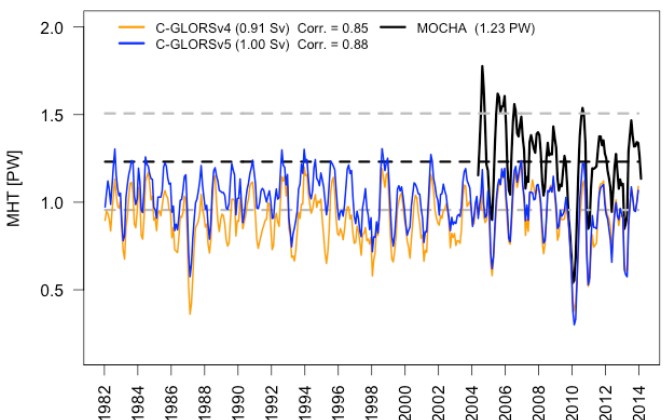

**Figure 9. Atlantic meridional overturning (AMOC, top panel, defined as maximum of the meridional streamfunction)
and heat transport (MHT, bottom panel) for C-GLORSv4, C-GLORSv5 and the RAPID array. Units are Sv (1 Sv =
1.E+6 m3/s) and PW (1 PW= 1.E+15 Watts), respectively. The plots also show in dashed lines the RAPID mean values
(black lines) +/- one standard deviations (grey lines).**



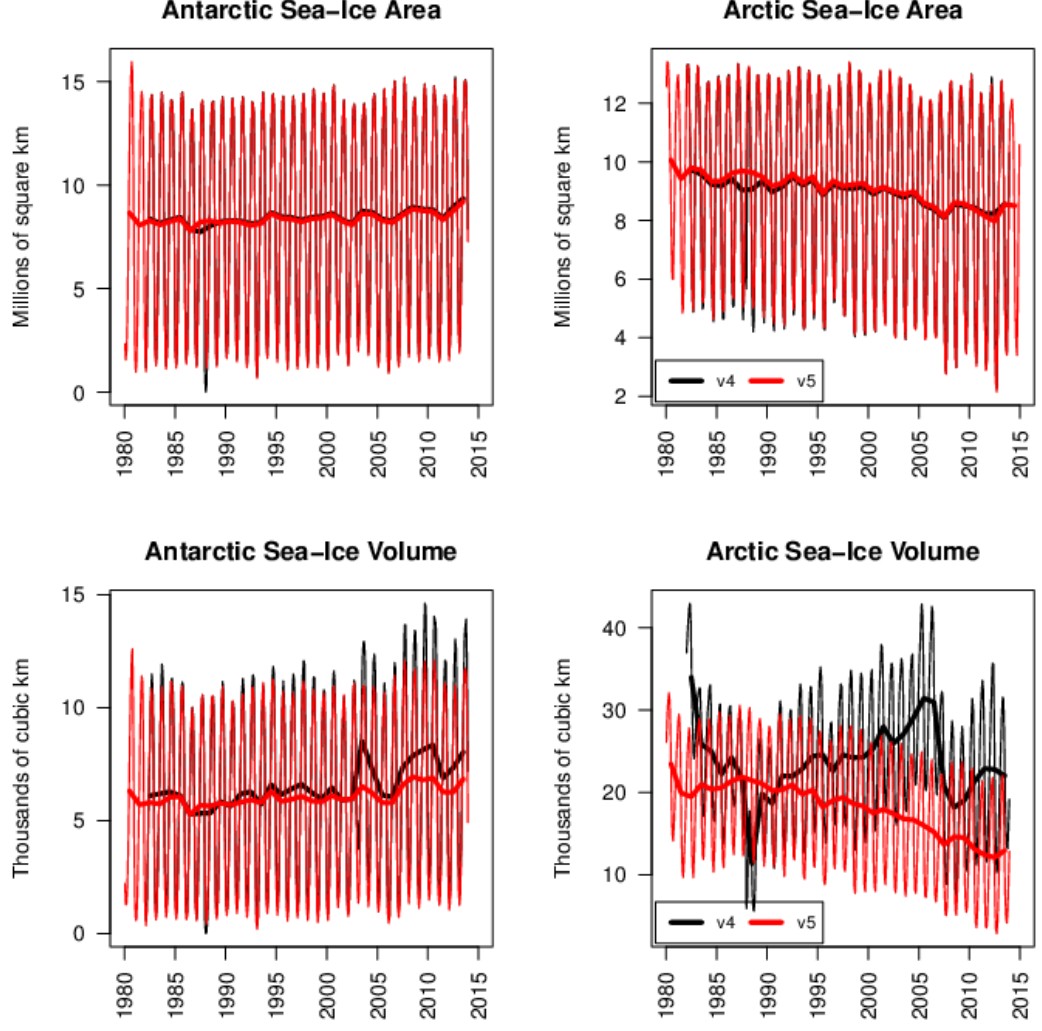

Figure 10. Antarctic and Arctic sea-ice area (top panels) and volume (bottom panels) in C-GLORS v4 and v5. Thin (thick) lines refer to monthly (yearly) means.




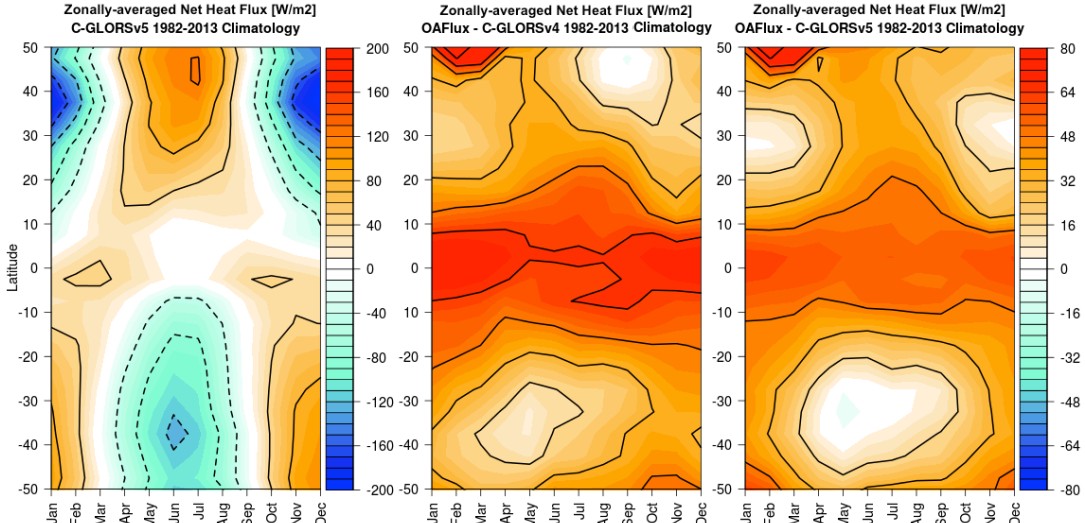

5    **Figure 11. Zonally averaged net heat flux (downwards) climatology (1982-2013): C-GLORS v5 (left panel), difference between OAFlux and C-GLORSv4 (middle panel) and difference between OAFlux and C-GLORSv5 (right panel).**