# Peer review of "C-GLORSv5: an improved multi-purpose global ocean eddypermitting physical reanalysis"

_Earth System Science Data, 2016_

## Referee Comment (RC1) · Anonymous Referee #1 · 28 Sep 2016

This article generates an updated version of a previous Global reanalysis data. Part of the derived products has been made available in NetCDF format at doi:10.1594/PANGAEA.857995. The method used in generating the data has a number of improvements in comparing with the one used in the old version.

As a global ocean reanalysis dataset, they can be useful in many areas. Methods and materials are mostly described in sufficient detail to support the data.

It is observed that there exist similar global reanalysis datasets, e.g., in in marine.copernicus.eu, ECCO or US HYCOM etc. Many components of the method description have also been published by the authors in other papers. However, these products haven't been inter-compared with the V5 data, either qualitatively or quanti-

tatively. This makes it difficult to evaluate the "state-of-the-art" and "uniqueness" of the data.

The new data (V5) have been validated and compared with the old data (V4), which shows improvements in simulating variability of the sea ice and AMOC. However some features of the products have been degraded. Overall T/S validation in Fig.5 shows that only water temperature in upper 80m in V5 has smaller RSME than V4, T/S in other layers for V5 are worse than V4. The global SST comparison also show significant signals in several areas of degradation (Fig. 6). These results have not been analyzed in the paper for details.

The comparison of C-GLORSv5 with C-GLORSv4 and observations has partly followed the common standards, e.g., SST, upper layer heat content, AMOC transport index etc. but not so comprehensive in terms of validation. In CMEMS, validation matrix and specific QUID reports are made for each dataset.

The length of the article is appropriate. Overall structure of the article is well defined and structured and readable. However, presentation can be further improved, including in figures, acronyms, symbols. Please refer to detailed comments.

Detailed comments Acronyms: There are many acronyms not given their full name when first time shown, eg EN3, EN4, MDT, SLA, SSH, DMSP, RMSE, PIOMAS, OAFlux, ISCCP etc.

P1, L10: "a state-of-the-art ocean reanalysis"

Reviewer: as the article does not perform any cross-validation with other reanalysis products from different systems, it is very hard to say it is a "a state-of-the-art ocean reanalysis".

P1, L18-19: "the new reanalysis outperforms the previous version, especially in representing the variability of global heat content and associated steric sea level, the upper ocean temperature and the thermohaline circulation."

Reviewer: the presentation can be more precise: "the thermohaline circulation" is AMOC, the "upper ocean temperature" is "slightly improvements in upper 80m but worse below for water temperature, and salinity is consistently worse in the upper 500m"

P1, L23: "a ocean" -> an ocean

P3, L29: "OcenVar" -> OceanVar

P4, L4: describe what is x and xb

P4, L10-15: VV and Vv shall make the same.

P6, L28: in the equation (5), ðİŻij has been used in equation (4), but are different parameters.

P6, L32: "cost" -> coast

P8, L20-21: the analysis on Fig.5 results should more precise and detailed, notifying that temperature is onyl slightly better in upper 80m but not 100m, maybe it's good to give a quantitative value of how much the temperature and salinity are better or worse in different levels.

P8, L28: while there is a 3.2% decrease of RMSE for global SST, there exist significant degrade in Gulf Stream, Kuroshio extension and circum-Antarctic ocean. It should also be mentioned is the accuracy of the NOAA SST, ie. about 0.6C.

P9, L9-10: "C-GLORV5 data start in 1980 unlike C-GLORSv4 (1982)" should be changed to "C-GLORV5 data start in 1980 unlike C-GLORSv4 in 1982"

P9, L10-15: it is suggested to use longer period for comparison the trend, e.g. 1982-2013 where NODC, V4 and V5 all have data. Use 2003-2011 for trend inter-comparison may be affected by the statistical significance due to very small number of samples.

P9, L18-25: Figure 8 is not convincing to show that V5 has a better heat content trend

than V4. Fig. 8c has 4 areas with large differences from V4, i.e., N. Atlantic including Gulf Stream, Kuroshio extension, South China Sea and circum-Antarctic ocean. Considering results from Fig.5 and Fig. 6, I would not conclude that these areas with significant differences are improved. The evaluation should reflect both significant signals of strength and weakness.

P9, L28: "RAPID-MOC" -> should this be RAPID-MOCHA?

P10, L5: 26N -> 26oN

P10, L21: "by a realistic" -> by a more realistic

P11, L3-4: "Based on this assessment, C-GLORSv5 proves a reliable tool for investigating the ocean and sea-ice interannual variability in polar regions."

Reviewer: I am not sure if this statement holds. In terms of ice volume, V5 has more reasonable results than V4. But it is hard to justify that V5 is a reliable tool for investigating the ocean and sea ice interannual variability in polar regions. Even PIOMAS is just another model. Regarding to real interannual variability in ice volume, one cannot say much due to lack of data.

P11, L7-9: it would be good to have a reference here.

P12, L10-12: the summary here on V5's quality on T/S should be more precise, and reflect results from Fig. 5 and Fig.6.

P22, Fig. 3: title of vertical axis is missing

P23, Fig. 4: the legend "180d": is this wrong? In the text it says 3 months, i.e., 90days.

P25, Fig. 6: the Caption should be rewritten

P26, Fig. 7: the correlation coefficients can be removed, as they are not explained and used in the text. Showing correlation generates a couple of issues: i) have the trends been removed before calculating the correlation? ii) the number of samples used and

significant level of correlation

P28, Fig. 9 lower panel: the legends need to be corrected, "Sv"->PW .

---

## Referee Comment (RC2) · Anonymous Referee #2 · 3 Oct 2016

**Review of ESSD-2016-38**
**C-GLORYSv5: an improved multi-purpose global ocean eddy-permitting physical reanalysis**
**Authors: Andrea Storto and Simona Masina**

October 3, 2016

**Recommendation**

Minor Revision

**Comments to Author**

This manuscript describes the latest version (v5) of the ocean re-analysis system C-GLORYS. The manuscript is well written, and while there are some things I would have presented somewhat differently, there is nothing I could suggest that would substantially alter the manuscript, and the authors should feel free to treat my minor comments below as mere suggestions.

My largest concerns were to do with the presentation and discussion with regards to the AMOC in Section 3.3. Firstly, it is somewhat difficult to make out the C-CGLORYS overturning underneath the RAPID observations in my paper version of Figure 9 – although I suppose that is one of the virtues of electronic media, as it is much easier to see in a zoomed in electronic version. It would also be illuminating, but not necessarily convenient, for the authors to show the non-Ekman component of the overturning streamfunction. All models, pretty much by definition, would replicate the Ekman component of the overturning derived by the RAPID observations, since that is solely determined by the wind stress forcing which is typically identical, or near identical to that used in the RAPID calculation. Replicating the density driven circulation, on the other hand, is more difficult — and ideally it would be the correlation between the non-Ekman portions of the overturning in the RAPID observations and C-GLORYS re-analysis that would be most interesting [Roberts et al., 2013]. In the absence of that calculation, however, it is noteworthy that neither C-GLORYS analysis appears to pick up the early period peak in the RAPID observations — although they do appear to pick up this 2005-2010 decrease in the RAPID overturning, and subsequent increase

after 2010 from about 2007 onward — which coincidentally would be when the ARGO float array is fully deployed, and their analysis potentially morphs from being largely driven by the SST nudging to one where the sub-surface profiles are playing a substantial role. Perhaps the authors may wish to substantiate on that further.

**Minor Comments**

1. p. 7, l. 2: 9% of the observations affected by the bug seems large. Are you inferring that 9% of the profiles were actually only surface measurements? Furthermore, is this 9% of the profiles, or 9% of all the profile observations at all levels?

2. Figure 3: There is a large spike in the monthly inflation value in 1993, very close to the coming on line of the altimeter data. Coincidence? Spurious?

3. p. 8, l 18: "namely the floats used represent a fairly independent dataset." Firstly, tacking this onto the end of the previous sentence does not make grammatical sense, but more importantly, you are being unduly brief with what in my mind is a fairly complex statement. What I believe you are saying is that because you are comparing observation minus background for floats (as opposed to say moored buoys), measurements at any given point can be consider independent, since (in principle) no one float makes repeat measurements at the same location. Perhaps it would be better to expand your statement somewhat – making it a complete sentence while you are at it.

4. Figure 4 and discussion p. 8, ll. 20-23. Only the global average observations minus background stats are shown. It would be worth at least showing the tropical (possibly Tropical Pacific) statistics that you note as significantly improved. Can this be attributed to the decrease in background-error standard deviation in the tropical Pacific. Conversely, observations minus background for the North Atlantic where the background-error has been increased and the skill decreased could be illuminating. Is the Gulf Stream more misplaced in the non-assimilated version of v5 compared to v4? Finally, you attribute the decreased skill at high latitudes to differing sea ice cover – but the sea ice cover should be largely constrained by the sea ice concentration observations. Is is not simply that you have increased the background error in this region as well? Note that the SST rmse is reduced at high latitudes as well.

5. Figure 8 and Section 3.2: The more accurate (compared to the NOAA/NODS estimates) trend in 2000m heat content in the Gulf Stream region seems at face value at odds with earlier statements regarding a loss of observation minus background skill in this region. However, there is also a decrease in SST rmse here as well, presumably due to the increased background error standard deviation – although the lack of flux correction could also play a role. Is the trend in 2000m heat content largely surface driven here?

6. Figure 9 and Section 3.3: As mentioned above, a comparison of the non-Ekman component of the Atlantic overturning streamfunction would be useful, but not essential.

7. Sections 2.2.5 and Section 3.4: Note on using PIOMAS as data. While PIOMAS does validate well with the sea ice thickness over the period it was validated – mostly ICESat data. However, it does not validate as well over data from more recent periods, possibly overestimating March ice thickness. However, I have no citable literature to back my claim, so this amounts to hearsay. Nevertheless, the Arctic ice volumes in C-GLORYSv5 are undoubtedly more realistic then those of v4. Undoubtedly, the spatial thickness patterns are close to those that are begin imposed by PIOMAS, nevertheless, a spatial map could be useful, especially if it can be compared with a satellite observations for a particular period. Laxon et al. [2003] could be used for an early altimeter based thickness estimate.

8. Figure 10: A yearly timeseries (with collapsed vertical axis) along with a seasonal cycle climatology might be more easily decipherable than the monthly timeseries shown. It might even be possible, with dual vertical axis, to plot area and volume on the same plot so that the number of sub-figures remains the same.

9. Section 3.4 and Figure 10. There is (presumably?) no ice thickness restoring performed in the Antarctic, yet the volume field in v5 is also more stable here than in v4. No mention was made of any sea ice improvements in the model, so can this be attributed to either the removal of the the flux corrections, or the changes (increase?) in background error covariance for the profiles. I note the error RMS error in SST is also reduced in both the Antarctic and Arctic. Would there have been similar improvement in the Arctic volume without the PIOMAS restoring?

**Typos and Grammatical Errors**

1. Title: I was always taught that a colon(:) should be followed by capitalization – and a complete sentence.

2. Section 2.1.1 Heading: Ocen $\rightarrow$ Ocean

3. p. 5, ll. 6-7: "constant value of 10m along the latitudes $\cdots$." I would have simply said "a globally uniform value of 10m $\cdots$."

4. p. 6, l. 15: "Differently" Use "Conversely"

5. p. 6, l. 30: Not really a typo, but insert $\alpha$ into the sentence to make this more explicit: "the threshold, $\alpha$, being increased from 6 to 9"

6. pp. 6-7, last and first line: You can't be shallower than the first level, but you can be shallower than the **middle** of the first level.

7. p. 7, l. 18: "passing" → "changing"

8. p. 7, l. 30: "which we found caused by" → "which we found was caused by"

**References**

S. Laxon, N. Peacock, and D. Smith. High interannual variability of sea ice thickness in the Arctic region. *Nature*, 425:947–950, 2003. doi: 10.1038/nature02050.

C. D. Roberts, J. Waters, K. A. Peterson, M. D. Palmer, G. D. McCarthy, E. Frajka-Williams, K. Haines, D. J. Lea, M. J. Martin, D. Storkey, E. W. Blockley, and H. Zuo. Atmosphere drives recent interannual variability of the atlantic meridional overturning circulation at 26.5n. *Geophysical Research Letters*, 40(19):5164–5170, 2013. ISSN 1944-8007. doi: 10.1002/grl.50930. URL `http://dx.doi.org/10.1002/grl.50930`.

---

## Author Comment (AC1) · 3 Nov 2016

**Response to Reviewer 1**

We thank both reviewers for the careful reading and suggestions on how to improve the manuscript, and in particular Reviewer 1 for pointing out ways to add more discussion in the analysis of results, further to many technical corrections. Below, we answer in details to all the points arisen from Reviewer 1. Sentences in italic are the reviewer's comments, while our reply is in bold, and includes the reply and the proposed modifications to the revised version of the manuscript.

*It is observed that there exist similar global reanalysis datasets, e.g., in in marine.copernicus.eu, ECCO or US HYCOM etc. Many components of the method description have also been published by the authors in other papers. However, these products haven't been inter-compared with the V5 data, either qualitatively or quantitatively. This makes it difficult to evaluate the "state-of-the-art" and "uniqueness" of the data.*

**We thank the reviewer for this comment. In the framework of the MyOcean project, our V4 reanalysis was extensively compared with the other three reanalyses produced in the framework of the project (Mercator-Ocean/GLORYS2, ECMWF/ORAP5, UniReading/UR4) and the main outcomes are included in Masina et al., (2015, Climate Dynamics). Within that intercomparison, it turned out the C-GLORSv4 is a state-of-the-art reanalysis for all parameters cross-compared (SST, temperature in the 0-800, and 0-2000 layers, sea-ice concentration, AMOC, volume transports in selected WOCE cross-sections) and also salinity content that however was not included in that publication. This cross-comparison allowed us to use C-GLORSv4 as the starting point to compare C-GLORSv5 with. Unfortunately, it is not straightforward and beyond the scope of this work an in-depth cross-comparison of our reanalysis with other global ocean reanalysis.**

**To account for this comment, we have modified the Introduction to explicitly reflect that C-GLORSv4 was already included in the MyOcean cross-comparison and proves a state-of-the-art reanalysis. We however have removed the reference to "state-of-the-art reanalysis" in the Abstract (replaced by "latest") to avoid any possible subjective definition.**

*The new data (V5) have been validated and compared with the old data (V4), which shows improvements in simulating variability of the sea ice and AMOC. However some features of the products have been degraded. Overall T/S validation in Fig.5 shows that only water temperature in upper 80m in V5 has smaller RSME than V4, T/S in other layers for V5 are worse than V4. The global SST comparison also show significant signals in several areas of degradation (Fig. 6). These results have not been analyzed in the paper for details.*

**We thank the reviewer for pointing this out and we modified part of the text in the Abstract, Results and Summary sections to explicitly state the metrics that get worse in V5 and justify them whenever is possible. Our**

experience suggests that when a consolidated product is upgraded, it is difficult to have all diagnostics that show improvements; nevertheless we believe that V5 contains important improvements and it is at the moment our recommended reanalysis product, which is also being widely used for a large number of applications and from many users worldwide.

*Detailed comments*

*Acronyms: There are many acronyms not given their full name when first time shown, eg EN3, EN4, MDT, SLA, SSH, DMSP, RMSE, PIOMAS, OAFlux, ISCCP etc.*

**Thanks for pointing this out; we have explicitly added the meanings of all acronyms.**

*P1, L10: "a state-of-the-art ocean reanalysis". Reviewer: as the article does not perform any cross-validation with other reanalysis products from different systems, it is very hard to say it is a "a state-of-the-art ocean reanalysis".*

**Please see the answer above to the general comment. The cross-comparison of v4 with other reanalyses (Masina et al., 2015, Climate Dynamics) is now explicitly mentioned, while we replaced "state-of-the-art" with "latest".**

*P1, L18-19: "the new reanalysis outperforms the previous version, especially in representing the variability of global heat content and associated steric sea level, the upper ocean temperature and the thermohaline circulation." Reviewer: the presentation can be more precise: "the thermohaline circulation" is AMOC, the "upper ocean temperature" is "slightly improvements in upper 80m but worse below for water temperature, and salinity is consistently worse in the upper 500m".*

**Please see the answer above to the general comment. Strengths and weaknesses of the new reanalysis have been precisely stated in the revised version of the manuscript, as the Reviewer suggests.**

*P1, L23: "a ocean" -> an ocean*

**Corrected**

*P3, L29: "OcenVar" -> OceanVar*

**Corrected**

*P4, L4: describe what is x and xb*

**We have added their description.**

*P4, L10-15: VV and Vv shall make the same.*

**Thanks for noting this; we have corrected it (all operators have lowercase index)**

*P6, L28: in the equation (5), $\delta^{.}IZ^{.}ij$ has been used in equation (4), but are different parameters.*

**Thanks for noting this; to avoid confusion the "threshold" of the background quality check is now indicated with the "gamma" Greek letter.**

*P6, L32: "cost" -> coast*

**Corrected**

*P8, L20-21: the analysis on Fig.5 results should more precise and detailed, notifying that temperature is only slightly better in upper 80m but not 100m, maybe it's good to give a quantitative value of how much the temperature and salinity are better or worse in different levels.*

**We thank the Reviewer for pointing this out: as mentioned in the answer to the general comment, we have precisely indicated the vertical range of improvement/worsening. According to Reviewer 2, we have also added panels on the skill score in the Tropical Pacific and North Atlantic Oceans.**

*P8, L28: while there is a 3.2% decrease of RMSE for global SST, there exist significant degrade in Gulf Stream, Kuroshio extension and circum-Antarctic ocean. It should also be mentioned is the accuracy of the NOAA SST, ie. about 0.6C.*

**We have added a discussion on the significant degradation in the regions mentioned by the Reviewer, which may be explained with the increased background-error variance in those areas (see also the Response to Reviewer 2). This would lead to heavier weights given to assimilated observations in the reanalysis, resulting in a worsening of the skill score in the validation against NOAA SST in areas with large variability. In the revised version, we also mention the accuracy of NOAA SST as suggested by Reviewer 1.**

*P9, L9-10: "C-GLORV5 data start in 1980 unlike C-GLORSv4 (1982)" should be changed to "C-GLORV5 data start in 1980 unlike C-GLORSv4 in 1982"*

**Corrected**

P9, L10-15: it is suggested to use longer period for comparison the trend, e.g. 1982- 2013 where NODC, V4 and V5 all have data. Use 2003-2011 for trend inter-comparison may be affected by the statistical significance due to very small number of samples.

**We have added also estimates for the trend within the period 1982-2013, for comparison with NODC estimates.**

P9, L18-25: Figure 8 is not convincing to show that V5 has a better heat content trend han V4. Fig. 8c has 4 areas with large differences from V4, i.e., N. Atlantic including Gulf Stream, Kuroshio extension, South China Sea and circum-Antarctic ocean. Considering results from Fig.5 and Fig. 6, I would not conclude that these areas with significant differences are improved. The evaluation should reflect

both significant sig- nals of strength and weakness.

**We have now better discussed the strengths and weaknesses: while skill score results seem worse in the Gulf Stream area, here the increase of heat content trend in the middle of North Atlantic gyre drives the sustained global signal in the last decade (see also the Response to Reviewer 2).**

*P9, L28: "RAPID-MOC" -> should this be RAPID-MOCHA?*

**We prefer to keep distinguished the sources for the volume transports and overturning (RAPID-MOC) and that for the meridional heat transport (RAPID-MOCHA), according to the fact that observational data belong to the two projects, respectively.**

*P10, L5: 26N -> 26oN*

**We have inserted the "degree symbol" in all occurrences**

*P10, L21: "by a realistic" -> by a more realistic*

**Corrected**

*P11, L3-4: "Based on this assessment, C-GLORSv5 proves a reliable tool for investigating the ocean and sea-ice interannual variability in polar regions." Reviewer: I am not sure if this statement holds. In terms of ice volume, V5 has more reasonable results than V4. But it is hard to justify that V5 is a reliable tool for investigating the ocean and sea ice interannual variability in polar regions. Even PIOMAS is just another model. Regarding to real interannual variability in ice volume, one cannot say much due to lack of data.*

**This has been shown by a recent study by Mayer et al. (2016, GRL) that makes use of C-GLORS data to investigate energy budget in the Arctic Ocean. We have softened the sentence and added the reference and a corresponding sentence. The new paragraph reads:**

Based on this assessment, C-GLORSv5 may be used as a tool for investigating the ocean and sea-ice interannual variability in polar regions. For instance, Mayer et al. (2016) make extensive use of C-GLORSv5 data (ocean heat content, sea-ice concentration and thickness, sea-ice velocities) to investigate the Arctic region energy imbalance.

*P11, L7-9: it would be good to have a reference here.*

**We have added the previous reanalysis paper (Storto et al., 2016a), and the intercomparison work (Valdivieso et al., 2015) that uses the same forcing configuration used in C-GLORSv4.**

*P12, L10-12: the summary here on V5's quality on T/S should be more precise, and reflect results from Fig. 5 and Fig.6.*

**Please see the answer above to the general comment. Strengths and**

weaknesses of the new reanalysis have been precisely stated in the revised version of the manuscript.

*P22, Fig. 3: title of vertical axis is missing*

**Added in the revised version.**

*P23, Fig. 4: the legend "180d": is this wrong? In the text it says 3 months, i.e., 90days.*

**Thanks for pointing this out, it was mistakenly reported as 180d, although it referred to 90d, i.e. 3 months. We have corrected the figure legend.**

*P25, Fig. 6: the Caption should be rewritten*

**We have rephrased the Caption: Map of differences of SST Root Mean Square error between C-GLORSv4 and C-GLORSv5. The RMSE is computed against NOAA SST ¼ daily analyses (Reynolds et al., 2007).**

*P26, Fig. 7: the correlation coefficients can be removed, as they are not explained and used in the text. Showing correlation generates a couple of issues: i) have the trends been removed before calculating the correlation? ii) the number of samples used and significant level of correlation*

**We thank the Reviewer for pointing this out: we have now mentioned the correlation in the text, reporting that the correlation is significant at 99% (bootstrapping) when greater than 0.25 for the 108 samples (monthly values from 2003 to 2011) considered here. We prefer to use this metrics rather than the correlation w.r.t. to NODC, because the latter provides only pentadal (5-year) means and it is strongly affected by the climatology during the first 10 years of the reanalysis period.**

*P28, Fig. 9 lower panel: the legends need to be corrected, "Sv"->PW .*

**Corrected**

---

## Author Comment (AC2) · 3 Nov 2016

**Response to Reviewer 2**

We thank both reviewers for the very careful reading and the many suggestions they proposed on how to improve the quality of the manuscript, and in particular Reviewer 2 for pointing out ways to better discus the AMOC validation, further to many technical corrections. Below, we answer in details to all the points arisen from Reviewer 2. Sentences in italic are the reviewer's comments, while our reply is in bold, and includes the reply and the proposed modifications to the revised version of the manuscript.

*My largest concerns were to do with the presentation and discussion with regards to the AMOC in Section 3.3. Firstly, it is somewhat difficult to make out the C-CGLORYS overturning underneath the RAPID observations in my paper version of Figure 9 – although I suppose that is one of the virtues of electronic media, as it is much easier to see in a zoomed in electronic version. It would also be illuminating, but not necessarily convenient, for the authors to show the non-Ekman component of the overturning streamfunction. All models, pretty much by definition, would replicate the Ekman component of the overturning derived by the RAPID observations, since that is solely determined by the wind stress forcing which is typically identical, or near identical to that used in the RAPID calculation. Replicating the density driven circulation, on the other hand, is more difficult — and ideally it would be the correlation between the non-Ekman portions of the overturning in the RAPID observations and C-GLORYS re-analysis that would be most interesting [Roberts et al., 2013]. In the absence of that calculation, however, it is noteworthy that neither C-GLORYS analysis appears to pick up the early period peak in the RAPID observations — although they do appear to pick up this 2005-2010 decrease in the RAPID overturning, and subsequent increase after 2010 from about 2007 onward — which coincidentally would be when the ARGO float array is fully deployed, and their analysis potentially morphs from being largely driven by the SST nudging to one where the sub-surface profiles are playing a substantial role. Perhaps the authors may wish to substantiate on that further.*

**We thank the reviewer for suggesting several ways to improve the discussions on C-GLORS with respect to the AMOC performances.**

**First, we certainly agree on the idea of evaluating the non-Ekman component between the reanalyses and RAPID in order to assess the density driven overturning circulation. To this end, we have evaluated the non-Ekman component, and we added a third (middle) panel in Figure 9 showing it (reported below). While there is a slight decrease in the correlation with RAPID, due as expected to the withholding of the Ekman transport, qualitative results on the larger correlation and increased mean value of the AMOC in v5 still hold.**

[Figure]

**Non-Ekman Atlantic Meridional Overturning**
**Maximum of the Meridional Streamfunction at 26N - Ekman transport**

Second, the reviewer questions why the reanalyses fail in capturing the first peaks occurring in the RAPID timeseries, which are due to the under-estimation of the western boundary current contribution (Florida Strait, FS, defined as the total transport at (80.1°W–77.4°W; 26.5°N)). The figure below reports the AMOC-FS (AMOC minus the transport across the Florida Strait), showing in particular that the peaks at the beginning of the RAPID time-series are not captured because of the FS under-estimation, due to poor observing network in 2005 along with the fact that the ¼ degree model resolution does not allow to perfectly resolving the Florida Strait geometry. Note that only the mismatch in the first two peaks of AMOC can be explained by the Florida transport under-estimation. We have explicitly added this explanation in the revised version of the manuscript.

[Figure]

**Non-Florida Atlantic Meridional Overturning**
**Maximum of the Meridional Streamfunction at 26N - FS transport**

1. *p. 7, l. 2: 9% of the observations affected by the bug seems large. Are you inferring that 9% of the profiles were actually only surface measurements? Furthermore, is this 9% of the profiles, or 9% of all the profile observations at all levels?*

   **We thank the reviewer for pointing this out: this was a typo as the percentage of mistakenly rejected surface in-situ observations is equal to 0.9 % on the average. We have corrected the value, also specifying that the number of mistakenly rejected surface in-situ observations is equal to about 3% at the beginning of the reanalysis period (ie at the beginning of the 1980s, when there are less profiles sampling deep waters).**

2. *Figure 3: There is a large spike in the monthly inflation value in 1993, very close to the coming on line of the altimeter data. Coincidence? Spurious?*

   **Thanks for pointing this out: although it is not straight-forward to prove, we also agree that the spike might depend on altimetry data ingestion, and suggested it as a possible cause in the revised version.**

3. *p. 8, l 18: "namely the floats used represent a fairly independent dataset." Firstly, tacking this onto the end of the previous sentence does not make grammatical sense, but more importantly, you are being unduly brief with what in my mind is a fairly complex statement. What I believe you are saying is that because you are comparing observation minus background for floats (as opposed to say moored buoys), measurements at any given point can be consider independent, since (in principle) no one float makes repeat measurements at the same location. Perhaps it would be better to expand your statement somewhat – making it a complete sentence while you are at it.*

   **The reviewer is correct in understanding what was implied by the sentence: we have reformulated the phrase to explicitly state the independence of floats observations given their spatial misplacement along time, and corrected the sentence.**

4. Figure 4 and discussion p. 8, ll. 20-23. Only the global average observations minus background stats are shown. It would be worth at least showing the tropical (possibly Tropical Pacific) statistics that you note as significantly improved. Can this be attributed to the decrease in background-error standard deviation in the tropical Pacific. Conversely, observations minus background for the North Atlantic where the background-error has been increased and the skill decreased could be illuminating. Is the Gulf Stream more misplaced in the non-assimilated version of v5 compared to v4? Finally, you attribute the decreased skill at high latitudes to differing sea ice cover – but the sea ice cover should be largely constrained by the sea ice concentration observations. Is not simply that you have increased the background error in this region as well? Note that the SST rmse is reduced at high latitudes as well.

**We have added a panel with the skill scores in the Tropical Pacific and North Atlantic oceans, showing the significant attenuation of Tropical biases in v5. A discussion on the reasons for different scores has been added to the revised version of the manuscript, as also suggested by Reviewer 1 (see also the Response to Reviewer 1). SST Skill scores in the Gulf Stream get worse in v5 mostly due to the increased background-error variances that also leads to a northward shift of the Gulf Stream separation. However, skill scores against in-situ profiles in the North Atlantic Ocean do not look worse than in v4. We have also better detailed the changes in scores at high latitudes, which may be due not only to the re-tuned background-error covariances but also to the use of uncorrected atmospheric forcing. In the first version, we meant that under sea-ice, NOAA SST data are extrapolated by sea-ice concentration data: SST analyses are affected by large uncertainties and the RMSE are not really "meaningful". We have however rephrased the sentence to make it clearer.**

5. *Figure 8 and Section 3.2: The more accurate (compared to the NOAA/NODS estimates) trend in 2000m heat content in the Gulf Stream region seems at face value at odds with earlier statements regarding a loss of observation minus background skill in this region. However, there is also a decrease in SST rmse here as well, presumably due to the increased background error standard deviation – although the lack of flux correction could also play a role. Is the trend in 2000m heat content largely surface driven here?*

   **We have now better discussed the strengths and weaknesses of the reanalysis, also according to Reviewer 1: while SST skill score results seem worse in the Gulf Stream area, here the increase of heat content trend in the central and eastern North Atlantic Ocean drives the sustained global signal in the last decade. The trend is largely driven by the surface.**

6. *Figure 9 and Section 3.3: As mentioned above, a comparison of the non-Ekman component of the Atlantic overturning streamfunction would be useful, but not essential.*

   **Please see the answer above to the general comment. We have added the comparison with the non-Ekman component.**

7. *Sections 2.2.5 and Section 3.4: Note on using PIOMAS as data. While PIOMAS does validate well with the sea ice thickness over the period it was validated – mostly ICESat data. However, it does not validate as well over data from more recent periods, possibly overestimating March ice thickness. However, I have no citable literature to back my claim, so this amounts to hearsay. Nevertheless, the Arctic ice volumes in C-GLORYSv5 are undoubtedly more realistic then those of v4. Undoubtedly, the spatial thickness patterns are close to those that are begin imposed by PIOMAS, nevertheless, a spatial map could be useful, especially if it can be compared with a satellite observations for a particular period. Laxon et al. [2003] could be used for an early altimeter based thickness estimate.*

We agree that PIOMAS is not a validating dataset, although it has been in turn extensively validated, as also the reviewer suggested. To this end we added also the reference to Schweiger et al. (2011, JGR). While we acknowledge that an independent dataset would be useful, that from Laxon is not accessible and there are copyright issues. We compared the 2004-2008 Winter mean thickness with ICESat (reported below), which highlights that the sea-ice accumulation in v4 is now fixed in v5. However we prefer not to include the figure, but mention this comparison in the text only.

[Figure]

8. *Figure 10: A yearly timeseries (with collapsed vertical axis) along with a seasonal cycle climatology might be more easily decipherable than the monthly timeseries shown. It might even be possible, with dual vertical axis, to plot area and volume on the same plot so that the number of sub-figures remains the same.*

We thank the reviewer for the suggestion and have replaced the Figure 10 in a Figure with 4 panels, 2 of which showing the yearly means, and 2 showing monthly climatology.

9. *Section 3.4 and Figure 10. There is (presumably?) no ice thickness restoring performed in the Antarctic, yet the volume field in v5 is also more stable here than in v4. No mention was made of any sea ice improvements in the model, so can this be attributed to either the removal of the flux corrections, or the changes (increase?) in background error covariance for the profiles. I note the error RMS error in SST is also reduced in both the Antarctic and Arctic. Would there have been similar improvement in the Arctic volume without the PIOMAS restoring?*

This is quite difficult to prove, but we believe that other factors such as retuning of the data assimilation system and use of uncorrected forcing might also contribute to a better sea-ice reconstruction. We explicitly added a sentence on this on the revised version.

*Typos and Grammatical Errors*

All the 8 Typos and grammatical errors indicated by Reviewer 2 have been corrected in the revised version of the manuscript.